# KOMPSAT-3 Digital Elevation Model Correction Based on Point-to-Surface Matching

**Hyoseong Lee [1,]\* and Michael Hahn [2]** 

[1] Department of Civil Engineering, Sunchon National University, Suncheon 57922, Korea
[2] Department of Geomatics, Computer Science and Mathematics, Stuttgart University of Applied Sciences, 70174 Stuttgart, Germany; Michael.Hahn@hft-stuttgart.de
\* Correspondence: hslee@scnu.ac.kr; Tel.: +82-61-750-3512

**Abstract:** In order to generate digital elevation models (DEMs) from high-resolution satellite images, the vendor-provided rational polynomial coefficients (RPCs) are commonly used. This results in a level of accuracy that can be improved by using ground control points (GCPs). The integration of the GCPs into the processing chain is associated with additional effort, since it requires the provision of GCPs as well as the measurement of its image coordinates. In this paper, the authors avoid the measurement of GCP image coordinates and propose a point-to-surface matching method to correct the DEM produced from KOMPSAT-3 satellite images and the provided RPCs. For point-to-surface matching, an existing network of GCPs was used in South Korea, the so-called united control points and the triangulation control points. Practical testing was summarized with the proposed method in which the root mean square error with respect to the horizontal position and the height reduced from 20 m and 6 m to 3 m and 2 m, respectively. This demonstrates that neither image coordinate measurements nor additional GCP point acquisition, e.g., by GPS, are necessary to convert a DEM generated from KOMPSAT-3 images and vendor-provided RPCs into a highly accurate DEM by using existing GCPs and point-to-surface matching.

**Keywords:** digital elevation models; rational polynomial coefficients; point-to-surface matching; united control points; triangulation control points

## 1. Introduction

The generation of a high-quality digital elevation model (DEM) using high-resolution satellite images requires rigorous sensor modelling or highly accurate approximations to the rigorous sensor model, such as rational polynomial coefficients (RPCs) including methods for the removal of exterior orientation biases [1–3]. The RPC approximation is popular mainly for its simplicity. For the KOMPSAT-3's images and RPCs that are used in the experimental investigations, the absolute accuracy specification is approximately 50 m with circular error at 90% probability (CE90) [4]. As demonstrated for many high-resolution imaging satellites using RPC georeferencing, there is also ample experimental evidence for KOMPSAT imagery that relative accuracy to meter level is attainable. The methods require one or more ground control points (GCPs). The provision of GCPs can be time consuming, especially if fieldwork such as GPS surveying is required. In most cases, the GCPs have to be measured in the images to make them usable for the methods.

The number of studies investigating bias compensation for RPCs both in image space and in object space is enormous, as discussed e.g., in [5]. The bias-correction approaches with shift, shift and drift, affine and the second-order polynomial models have been investigated with Ikonos, Quickbird, Worldview, GeoEye and other high-resolution imageries. The best performance was usually achieved with the affine transformation model [5].

The wide desire of research to metrically exploit the high resolution satellite imagery to the maximum extent possible has led to early developments of rigorous mathematical models for orientation and triangulation. Under ideal conditions of high-quality ground control, precise image measurements and the provision of sensor calibration data, ground point determination to 0.3 pixels is possible [6]. In practical tests, accuracies of between 0.5 and 2 pixels have more commonly been encountered.

Based on rigorous models, concepts are being pursued without relying on conventional GCPs [7–9]. Bouillon et al. [7] performed bundle block adjustments using tie points without GCPs to improve the quality of DEM generation from SPOT-5 HRS stereo images. The horizontal and vertical DEM accuracy for 90% of the points was 8.4 m and 4.5 m, respectively. The model of Toutin et al. [8] with virtual control points has led to elevation errors with 2.6 m and 2.1 m (68% confidence level) for World-View-1 and 2 stereo images. In an experiment with a huge block of 26.406 ZY-3 images covering 93% of China's mainland, Yang et al. [9] achieved a positioning accuracy with calibrated RPCs of 15 m and improved it by bundle adjustment with virtual control points to 5 m.

Other approaches to improve the quality of generated DEMs have their origin in point determination using DEMs as control information [10,11]. Originally, the six parameters of absolute orientation of photogrammetric stereo models had to be determined using the given digital elevation models. The estimation of the transformation parameters between points and an elevation model presents itself as a matching problem. The determination of the parameters and the determination of the corresponding points on the terrain surface are part of the solution of the matching problem. Given that the orientation difference between the points and the surface is small, the temporary pairing of the points with points on the surface is based on the same horizontal position. By undertaking pairing and minimization of the height differences iteratively, every iteration brings the points closer to the surface. The method is also called the least-height difference (LHD) algorithm because the sum of the squared height differences is minimized to estimate the transformation parameters. Lidar data triangulated to surface models may replace a DEM in LHD. Likewise, two elevation models can be used as input for the LHD algorithm. An extension of LHD to 3D surface matching for 3D object registration can be found in [12].

In [13], a shuttle radar topography mission DEM is used for RPC correction of images from TianHui-1 satellite based on the LHD algorithm. Kim and Jeong [14] examined the suitability of the LHD algorithm for precise mapping of pushbroom images. They emphasized that the 3D similarity transformation is evident when the errors occur only in the form of the time-invariant position and attitude biases of the coordinate frame of the pushbroom images.

This study proposes a point-to-surface matching technique in which the sum of squared distances of points to a surface is minimized to estimate the transformation parameters between the points and the surface. Our goal is to correct the DEM generated by this study from KOMPSAT 3 stereo images and the included RPCs. A bilinear surface model is used to describe the DEM. The points used for matching are a subset of the about 4000 national unified control points (UCPs) and approximately 17,000 triangulation control points (TCPs) that have been established all over South Korea by the Korean National Geographic Information Institute (NGII). The UCPs result from geodetic measurements in which GNSS and gravimetric measurements are included in the geodetic network processing. The TCPs have their origin in triangulation-based measurements, which today are supplemented by GNSS measurements. These ground control points are usually not visible in a KOMPSAT image and therefore their image coordinates cannot be measured. With the national control point record system [15], the 3D ground coordinates of UCPs and TCPs can be easily downloaded. Section 2 describes the model of the point-to-surface matching and presents a flowchart of the study. Experimental investigations on the DEM correction of a Kompsat-3 stereo image are presented and discussed in Section 3. Illustrated are the algorithmic aspects of the iterative solution. The achieved accuracies are verified by ground truth measurements.

## 2. Point-to-Surface Matching Model

A starting point for the formulation of the point-to-surface matching problem is the task to be solved. One data set is a DEM that is generated from a high-resolution satellite image pair using the vendor-provided RPCs. It is well known that due to the inherent sensor orientation bias errors, the generated DEM is distorted. The second data set are the UCPs and TCPs, i.e., an existing network of highly accurate control points maintained by NGII mainly for surveying purposes. These points are given by 3D coordinates and they lie on the terrain surface. The points and the surface are to be set in relation to each other by a transformation. This study follows the argumentation of [14] and approximates the transformation by a 3D similarity transformation which implies the time-invariant position and attitude biases of the coordinate frame of the pushbroom images.

The objective of the matching approach is to find the transformation between the points and the surface, which consists of a 3D rotation and a 3D translation as well as a scale parameter, such that the sum of the squared distances between the points and surface is minimized. For the LHD algorithm, the height difference between a point $p_i$ and a point $q_i^*$ on the DEM is used as the measure of distance. Instead of the height difference, this study's approach shall use the shortest distance $d_i$ from the point $p_i$ to the surface at point $q_i$ as shown in Figure 1.

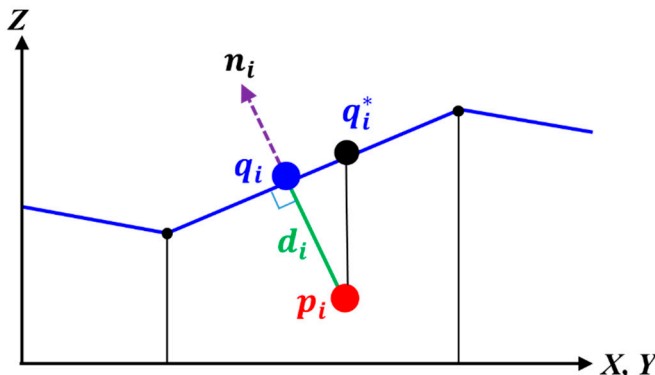

**Figure 1.** Distance measure of point-to-surface matching between ground control points (GCPs) ($p_i$) and surface Z(X,Y) of a digital elevation models (DEM).

The determination of the shortest distance $d_i$ from point $p_i$ to the surface can be formulated as an optimization problem of finding the corresponding point $q_i$. If the surface is described by a plane (as sketched in Figure 1), this is straight forward as any point on the plane (e.g., also $q_i^*$) can be used to calculate the distance $d_i$ of point $p_i$ from the plane according to:

$$d_i = n_i^T(p_i - q_i) \tag{1}$$

where $i$ indicates the i-th point, $n_i$ is a normal vector of length 1, $p_i$ is the location of the GCP, and $q_i$ is the location of the corresponding point on the surface. For non-planar surfaces patches like bilinear surfaces (Figure 2), the optimization can be solved iteratively with approximating tangential planes, for which the surface point that best matches $q_i$ can be found iteratively.

The 3D similarity transformation model reads as follows:

$$p_i^* = sR\,p_i + t \tag{2}$$

$p_i^* = q_i + \Delta_i$ is the location of the transformed GCP, $R = R(\omega, \varphi, \kappa)$ is the 3D rotation matrix, $t$ is a 3D translation vector and s is a scale parameter. Ideally, the transformed GCPs $p_i^*$ lie on the surface, so that the discrepancies $\Delta_i$ disappear. Since the scale has proved to be superfluous in the experimental investigations, it is left out of the equation at this point.

Assuming the DEM is modelled implicitly by $F(X, Y, Z) = 0$, the surface can be linearized by Taylor series expansion at any surface point $\left(X^0, Y^0, Z^0\right)$ according to:

$$F(X, Y, Z) = F\left(X^0, Y^0, Z^0\right) + \frac{\partial F(..)^0}{\partial X} dX + \frac{\partial F(..)^0}{\partial Y} dY + \frac{\partial F(..)^0}{\partial Z} dZ \tag{3}$$

where the first derivatives ($\frac{\partial F(..)^0}{\partial X}, \ldots$) are the components of the surface normal $n_0$ at the considered surface point. According to Equation 3, all points $q$ with $(X, Y, Z)$ coordinates are on the surface $F(X, Y, Z)$. If the local neighborhood of a surface point $q_0$ is considered, Equation (3) leads to the approximation of the surface by a tangential plane $n_0^T(q - q_0) = 0$ at the surface point.

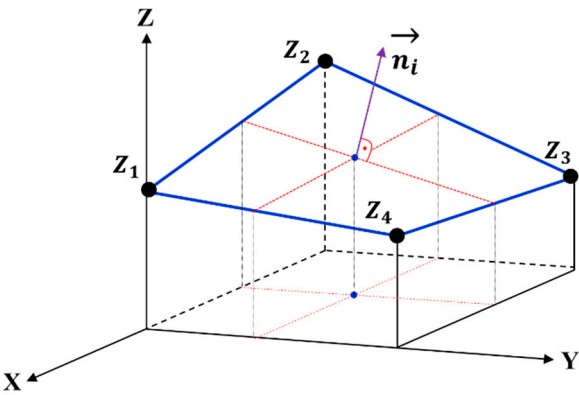

**Figure 2.** Normal vector of the tangential plane at a certain location of a bilinear surface.

The least squares approach for point-to-surface matching follows directly by taking points outside the surface (in this case, the GCPs) into account. The GCPs $p_i$ substitute the surface points $q$ and $q_i$ replaces $q_0$. If, in addition, the surface normal $n_0$ is normalized to length 1 ($n_0 / \|n_0\|$), Equation (1) is obtained that calculates the signed Euclidean distance to the tangential plane.

For the least squares approach, suitable approximate values that are available for the parameters of the geometric transformation are required. In matching of GCPs to a KOMPSAT-3 DEM, it can be assumed that a rotation matrix $R = E$ and a translation vector $t = 0$ are suitable starting values. Using the notation $p_i^*$ for iteratively transformed GCPs:

$$d_i = n_i^T\left(p_i^* + dp_i^* - q_i\right) \text{ with } dp_i^* = dR\, p_i + dt. \tag{4}$$

The differential term is described in more detail as follows:

$$\begin{pmatrix} dx \\ dy \\ dz \end{pmatrix} = \begin{pmatrix} a_{11} & a_{12} & a_{13} \\ a_{21} & a_{22} & a_{23} \\ a_{31} & a_{32} & a_{33} \end{pmatrix} \begin{pmatrix} d\omega \\ d\varphi \\ d\kappa \end{pmatrix} + \begin{pmatrix} dt_x \\ dt_y \\ dt_z \end{pmatrix} \tag{5}$$

where the coefficient terms $a_{ij}$ are calculated by the partial derivatives with respect to the three rotation angles ($\omega, \varphi, \kappa$).

The linearized model of Equation (4) together with Equation (5) leads to the matrix notation $d = Ax - l$, where $A$ is the design matrix, $x = \left(d\omega, d\varphi, d\kappa, dt_x, dt_y, dt_z\right)^T$ the vector with the geometric transformation parameters and $l = \left(\ldots, n_i^T\left(p_i^* - q_i\right), \ldots\right)^T$ represents the observation vector.

Minimizing the sum of the squared distances $e = \sum_{i=1}^{N} d_i^2 = min$ results in the standard least squares estimates of the parameters $\hat{x} = \left(A^T P A\right)^{-1} A^T P l$. The weights are determined in an appropriate manner. In our experiments, equally weighted observations $P = E$ are assumed.

The standard formulation of the estimation process can be extended as discussed e.g., in [12,16]. An extension of the observation vector by pseudo observations of the unknown transformations parameters [12] gives additional control over the unknown. In [16], two surfaces each with their own error behaviour, are treated in a suitably developed estimation process. A corresponding application to the matching of DEM and GCPs considers both inputs as stochastic quantities in the adjustment. The fact that the matching procedure with its iterative parameter estimation can be classified as one of many ICP solutions shows [17].

The iteration process of point-to-surface matching is then as follows:

(1) At iteration step 0, $R$ and $t$ are defined by a unit matrix and a zero vector, so that the transformed point $p_i^*$ is identical to $p_i$.

(2) With the horizontal coordinates of point $p_i^*$ a surface point ($q_i^*$ in Figure 1) of the DEM is interpolated and the tangential plane with the unit normal vector $n_i$ is calculated at this point according to $d_i = n_i^T(p_i^* - q_i^*)$. An improved location for the corresponding surface point is determined by $q_i = p_i^* + d_i n_i^T$ and the normal vector is updated at $q_i$. The iterative calculation of $(d_i, q_i)$ is repeated (mostly 2 times) until $\|q_i^k - q_i^{k-1}\| < \epsilon_q$. Since the DEM from KOMPSAT-3 is smooth an $\epsilon_q$ of 1 m was sufficient.

(3) The linearized equation system according to Equations (4) and (5) is set up and solved by least squares.

(4) The rotation and translation parameters are updated iteratively until convergence is achieved. The iterations are terminated as soon as the estimated corrections of the transformation parameters are below a certain limit $\|\hat{x}\| < \epsilon_{\hat{x}}$. In the implementation, $\epsilon_{\hat{x}}$ was divided into a threshold for the translations $\epsilon_t = 1$ cm and a threshold for the rotations $\epsilon_r = 10^{-4}$ deg.

If the horizontal displacement between the DEM and the GCPs is greater than the DEM grid width, the iterative estimation of the parameters causes the transformed GCPs to move across adjacent grid cells. The normal vector may differ significantly between adjacent DEM grid cells. An experimentally observed consequence was that the estimates were obtained that transform GCPs into adjacent grid cells during one iteration and transforms them back in the following iteration. In order to mitigate this influence, only one third of the iteratively estimated values ($\hat{x}/3$) were taken into account when updating the parameters. This smoothes the iteration behavior within the least-squares estimation process but causes an increase in the number of iterations. A more favourable iteration behaviour would probably be achieved with the Levenberg-Marquardt algorithm [18].

Figure 3 gives an overview of the processing steps underlying the experimental investigations.

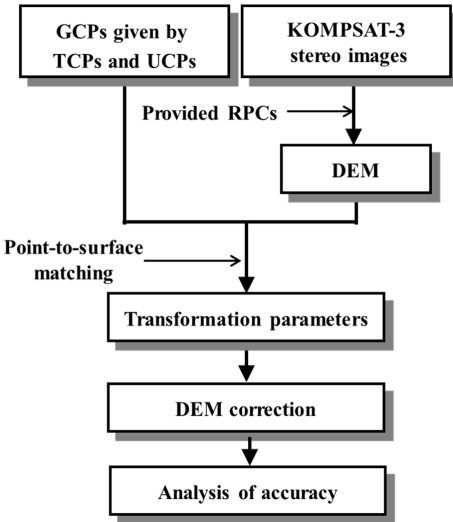

**Figure 3.** Flow chart for the correction of the DEM using point-to-surface matching.

A DEM was generated from the KOMPSAT image pair using the vendor-provided RPCs. The 3D coordinates of the GCPs (TCP and UCP) for the area covered by the DEM were downloaded from the NGII website [15]. The proposed point-to-surface matching was used to determine the transformation parameters between the GCPs and the DEM. Since the number of GCPs is comparatively small, matching is not very time-consuming even with many iterations. With the inverse transformation parameters, the DEM was corrected. For the accuracy investigations, GPS points were collected. They serve as ground truth for the accuracy assessment.

## 3. Experimental Investigations, Results and Discussions

The test site for the experiments is shown in Figure 4. The KOMPSAT-3 images cover an area of approximately 20 km by 16 km in the Yangsan city region. The very mountainous area is forested at higher altitudes. Table 1 shows the details of the panchromatic KOMPSAT-3 image pair. The ground resolution of the two images is 85 cm and 86 cm, respectively. Roll and yaw angles of both images point in a similar direction. The pitch angles differ essentially in their sign.

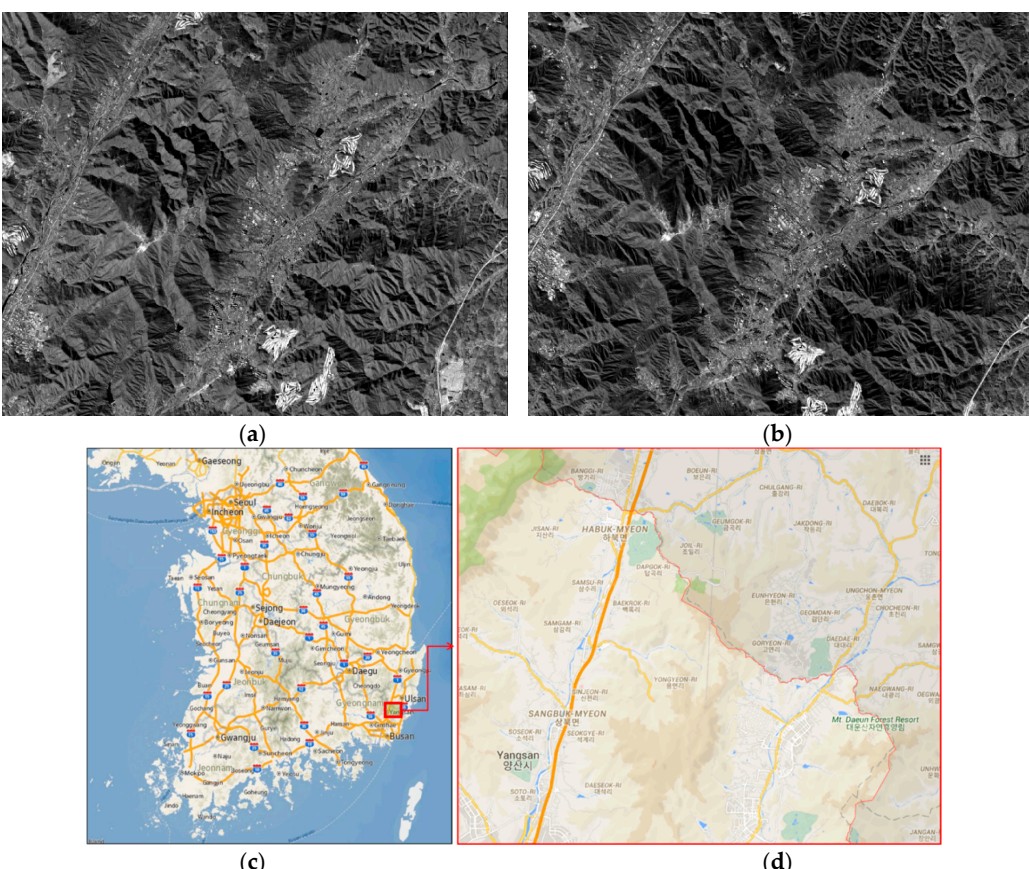

(a) (b)

(c) (d)

**Figure 4.** Study area: (**a**,**b**) KOMPSAT-3 stereo pair for Yangsan, (**c**,**d**) map of the Korean peninsula with the Yangsan city region in South Korea.

**Table 1.** Specifications of the satellite images used.

|  | First Image | Second Image |
| --- | --- | --- |
| Acquisition date/time | 2013-01-25/04-19-49 | 2013-01-25/04-21-07 |
| Image size (pixels) | 24,060 × 18,304 | 24,060 × 18,792 |
| GSD | 0.86 m | 0.85 m |
| Roll angle | 18.64° | 16.50° |
| Pitch angle | 19.85° | −20.16° |
| Yaw angle | −2.32° | −3.88° |

All national UCPs and TCPs located in the project area are visualised in Figure 5.

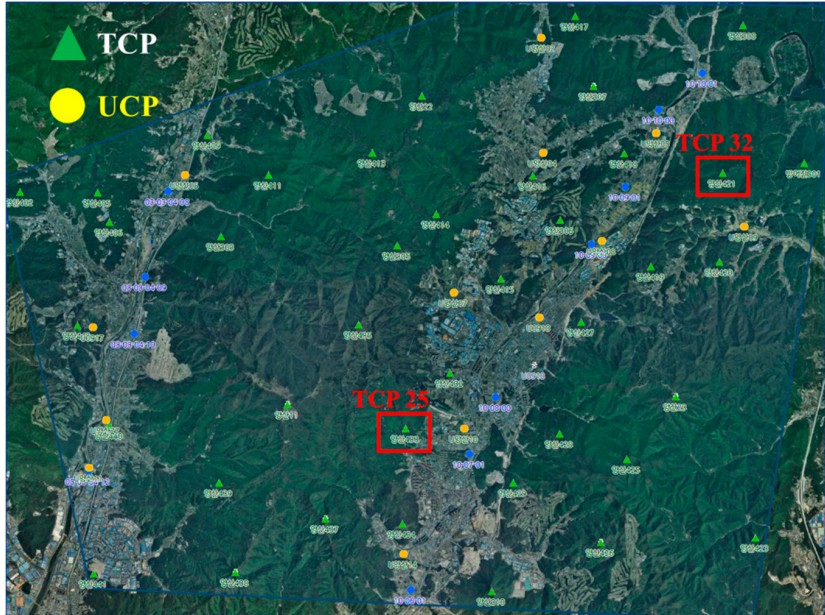

**Figure 5.** Distribution of unified control points (UCPs) (yellow dots) and triangulation control points (TCPs) (green triangles) in the Yangsan project region as shown on the NGII website. The two highlighted TCPs (#25 and #32) are discussed in more detail in the analysis of the results.

A total of 13 UCPs and 40 TCPs were available for the point-to-surface matching investigations. The figure shows that most of the UCPs are located in low-elevation areas of a surrounding urban area, while most of the TCPs are in exposed locations in the mountains. The UCPs in the test area were mostly created after 2012 and are monitored annually. Most of the TCPs have been available since 1998; they were verified and updated from May 2012 to June 2013. Due to the temporal proximity to the recording of the KOMPSAT-3 images, it can be expected that the images and the GCPs should fit together well. ERDAS IMAGINE was used to generate a DEM with 5 m grid spacing using the KOMPSAT-3 image pair and the vendor-provided RPCs (Figure 6).

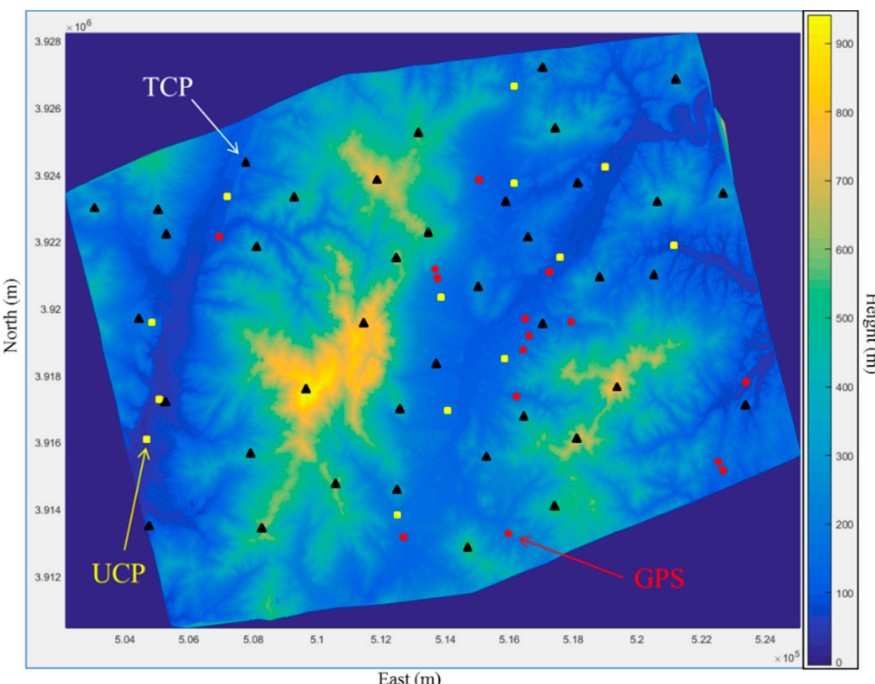

**Figure 6.** Generated DEM for the Yangsan region. UCPs, TCPs and GPS check points are overlaid.

For ground truth investigations, 15 checkpoints were recorded by a GPS survey in April 2013 within the region of the DEM (red points in Figure 6). These points are well recognizable in the KOMPSAT-3 image pair. By measuring the image coordinates with sub-pixel accuracy and spatial intersection using the RPCs of the image pair, the ground coordinates of these points were determined. The coordinate differences to the GPS ground truth coordinates are plotted in Figure 7.

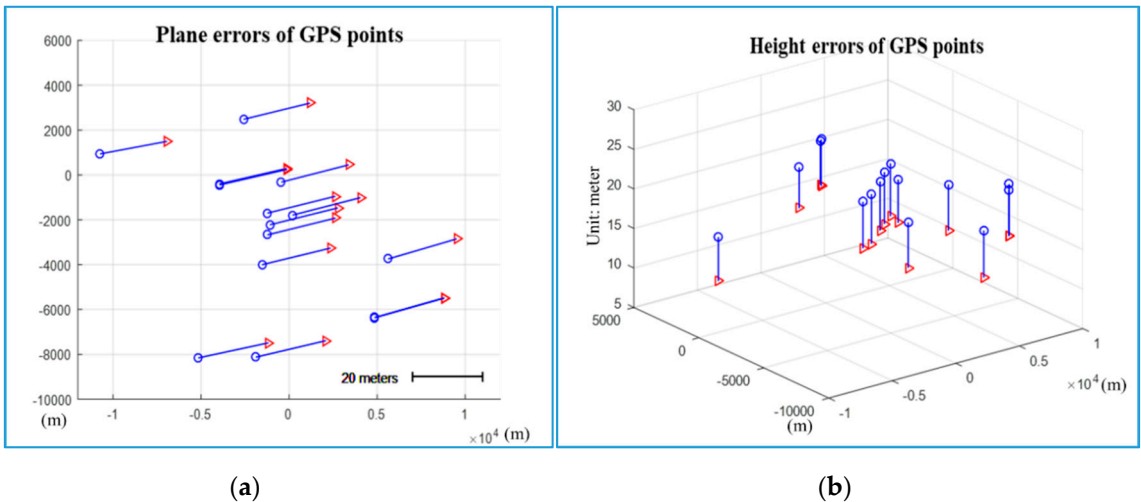

(**a**)                                                          (**b**)

**Figure 7.** Deviations in GPS checkpoints. Displayed are (**a**) the horizontal and (**b**) the vertical differences between measured GPS coordinates and coordinates reconstructed from the KOMPSAT image pair (circles: reconstructed positions; triangles: GPS positions).

These differences serve as a reference for the matching investigations. The GPS measurement accuracy was below 5 cm, the accuracy of the stereo measurements of the image coordinates of the GCPs between 0.5 and 0.7 pixels, equalling a horizontal and vertical accuracy of 40 to 60 cm. Overall, the differences in all three coordinate axes are at an accuracy level of half a meter.

In the experiment, the authors used the GCPs to apply the point-to-surface matching with the following two cases of transformation parameters:

Case 1: Three translation parameters: $(X_t, \ Y_t, Z_t)$

Case 2: Three rotation angles and three translation parameters: $(\omega, \varphi, \kappa, X_t, \ Y_t, Z_t)$

First, the iteration behaviour of the matching procedure in both cases is investigated. Here, the interest is in the horizontal and vertical displacements (translations), their changes from iteration to iteration and the sum of squared distances that are visualized by a mean distance of the points to the tangential planes $\overline{d} = \sqrt{\sum_{i=1}^{N} d_i^2 / (N-6)}$. This study also considers how the normal vectors $\boldsymbol{n}_i$ change at the surface points $\boldsymbol{q}_i^k$ during the iterations k.

*3.1. Observed Iteration Behaviour*

Figure 8 illustrates the iteration behaviour for both cases. The first row of the figure shows the shifts and rotation angles resulting from the estimation process that are updated at each iteration. The behaviour of the angle $\kappa$ is particularly noticeable, which increases steadily in the first 5 iterations and then returns to almost its initial value in a further 15 iterations. The estimated parameters themselves, i.e., the estimated shift differences and rotation differences, are plotted in the second row of the figure. As long as the estimation process causes the transformed points to fall into new DEM grid cells, the estimated shift differences of up to 3 meters are visible in the figures of both cases (Figure 8, 2nd row). The shift corrections larger than the grid size of the DEM would cause all transformed points to be in new grid cells in the next iteration. In case 1, the estimates for the 3 D shift remain unchanged (less than 1 cm changes) after 15 iterations. Figure 8 (3rd row) shows the convergence of the process in which the mean distance of the transformed points to the surface reduces

from 6.5 m at the beginning to less than 3 m at the end of the iterations. Adding the rotation angles to the geometric transformation (case 2) requires a few more iterations until convergence of the algorithm is reached. While the rotation angles show a less targeted iteration pattern, the iteratively estimated shifts are broadly similar in both cases.

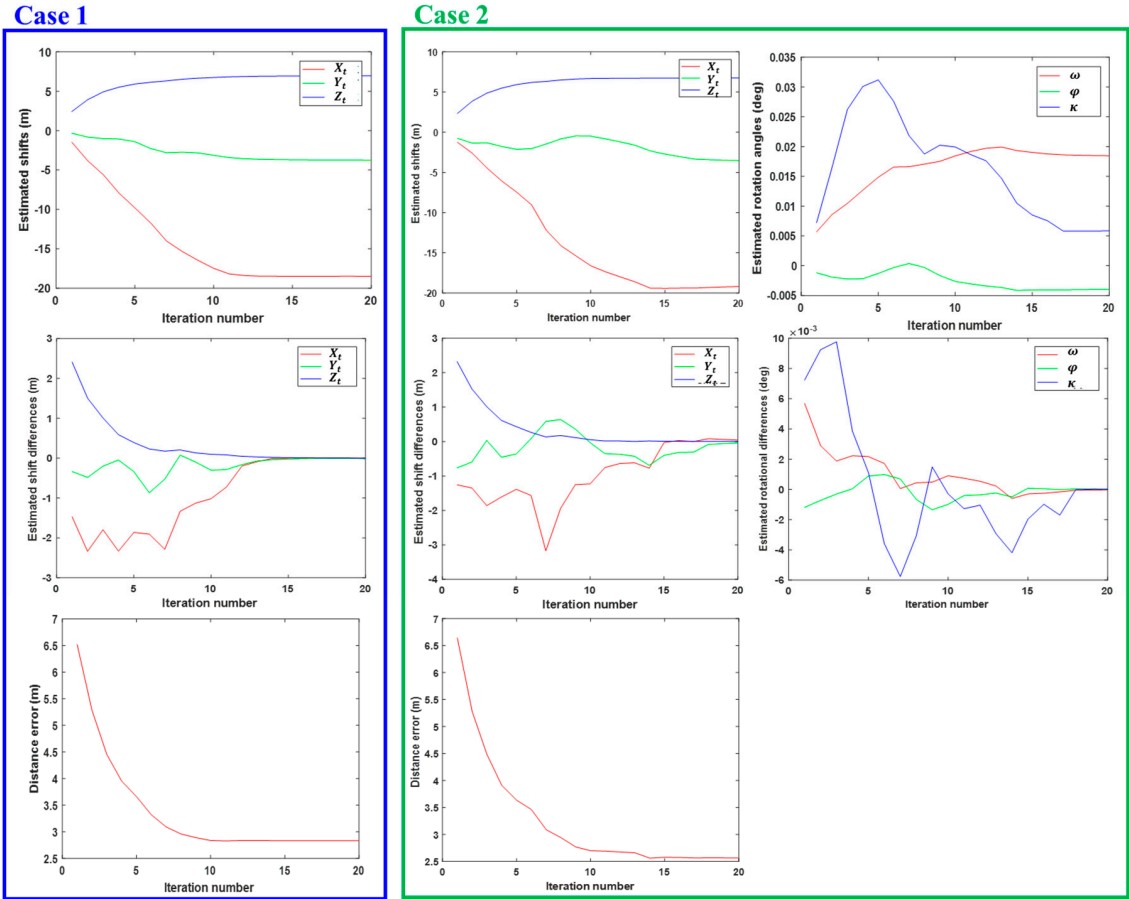

**Figure 8.** Convergence behaviour of the matching algorithm. The left and right columns illustrate the estimation process in case 1 (three translation parameters) and case 2 (three rotation angles and three translation parameters), respectively.

In order to observe the changes of the normal vectors $n_i$ in the course of the iterations, the components $(n_x, n_y)$ at the surface points $q_i^k$ are visualised for some iteration steps $k$. $(n_x, n_y)$ are gradient vectors in $x$ and $y$ taken from the unit normal vectors. Figure 9 shows these gradient vectors for six selected iteration steps. Case 2 is used as an example. During the first 15 iterations, changes in the direction and slope (indicated by the length of the vectors) can be observed at many surface points. After that, only small changes are visible at a few points and from iteration 18, no change of the gradient vectors is visible anymore.

The comparison of the points with larger slopes with the representation of the TCP and UCP in Figures 4 and 5 makes it clear that in particular the TCP, which are located in the upper regions of the mountains, are responsible for this. Figure 10 shows the slope angles at the surface points that correspond to the TCPs and the UCPs. It is clearly visible how strongly the inclination angles differ between the terrain locations of the TCPs and the UCPs. The larger gradient vectors contribute in particular to the determination of the horizontal shift. We will come back to Figure 10 later to discuss the relationship of slopes and calculated distances between GCPs and DEM before and after matching (Figure 10, right diagram).

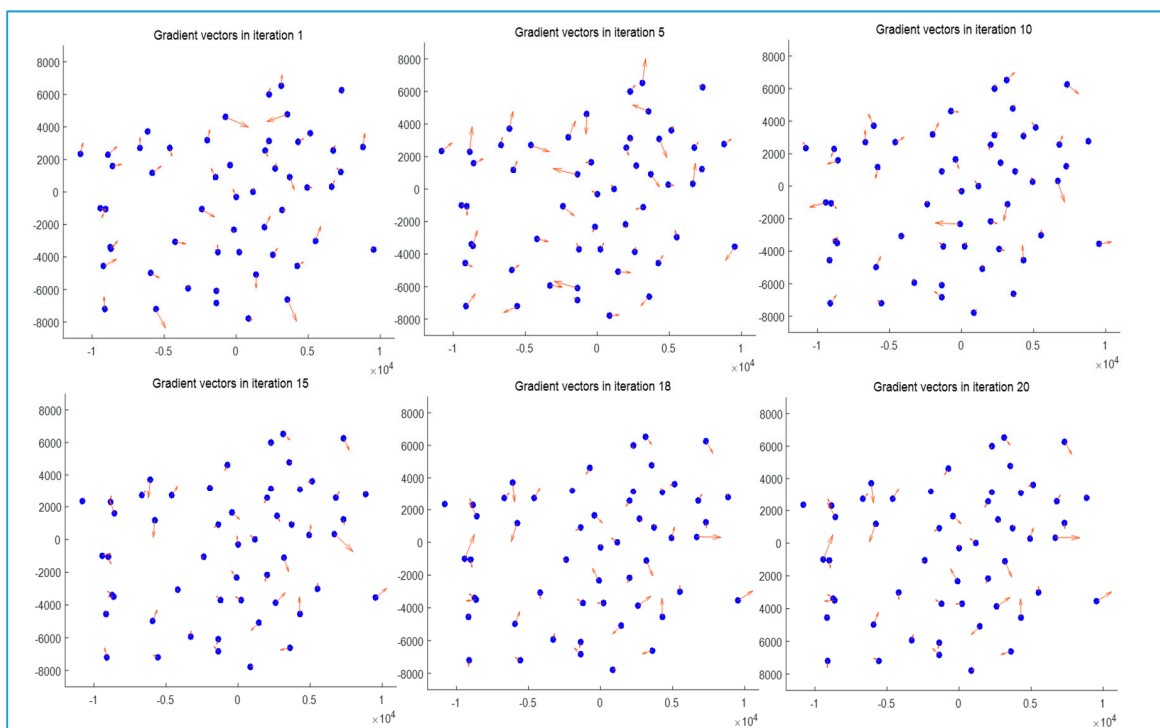

**Figure 9.** Gradient vectors in surface points $q_i^k$ for some iteration steps (the unit of axes is meters).

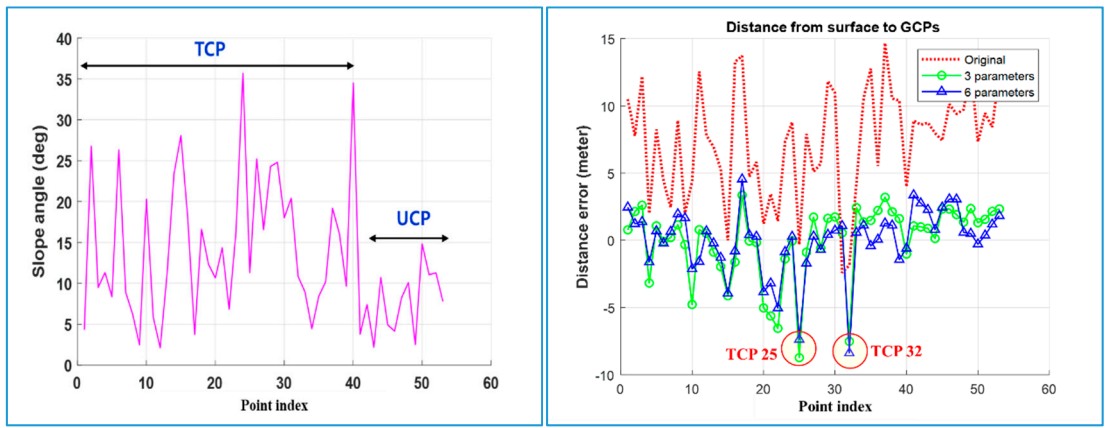

**Figure 10.** Slope angles at the surface points (left) and distances between GCPs and DEM before and after matching (right).

The iteration behaviour related to the transformed GCPs (Equation 2) is illustrated in Figure 11 for case 2. For this purpose, the vector field is calculated, which develops from the transformation of the GCPs with respect to their initial position. The horizontal components are visualised for some iterations as well as the vertical component, but that only at the final iteration step. While in the first iterations, a rotational influence can be recognized, the translational component dominates at the end of the iterations. The horizontal and vertical movements of the transformed GCPs from the first iteration to the last iteration are approximately 20 m horizontally and 6 m vertically. As expected, this fits well to the coordinate differences that result from the KOMPSAT image measurement and the GPS recordings (cf. Figure 7). The accuracy investigation for this follows in the next section.

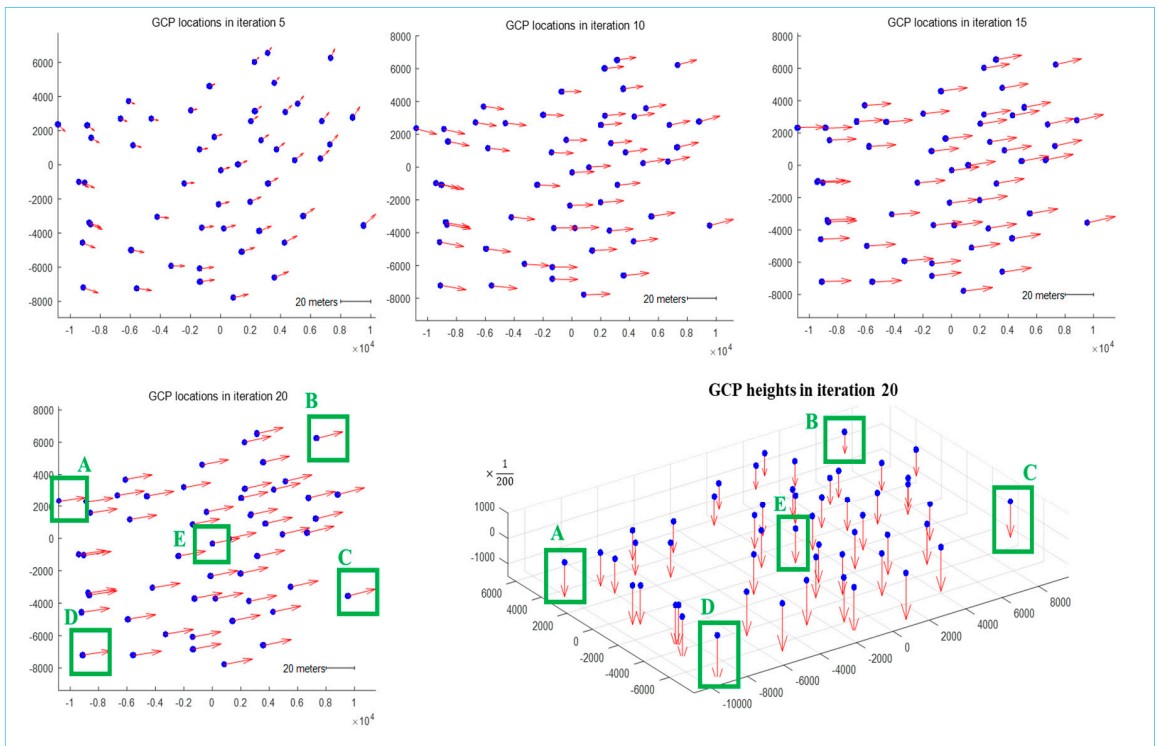

**Figure 11.** The iteration behaviour related to the transformed GCPs. Visualised is the vector field $\left(p_i^* - p_i\right) \forall \, i = 1, N$ for selected iterations (the unit of the axes is meters).

The transformation sequence over 20 iterations is illustrated in Figure 12 exemplarily for five GCPs marked by the green border in Figure 11. Four GCPs were selected which are close to the respective corners of the project area as well as one GCP which is located in the middle. The left column in Figure 12 shows the horizontal movement of the points over all iteration steps. In the right column, the vertical movement is visualized in a 3D view. The movement of GCPs B and E over all iteration steps appear remarkably smooth, which is particularly visible in the 2D plot (left column of Figure 12). Essentially, a smooth movement pattern can also be observed with the other three GCPs, although the 2D track appears more curved. With all GCPs, large movements of the points occur, more or less, during the first 15 iteration steps. The incremental displacements then become smaller and smaller until they are virtually zero after 18 iterations. As expected, the movement of the GCP correlates with the convergence behaviour of the estimated parameters (cf. Figure 8).

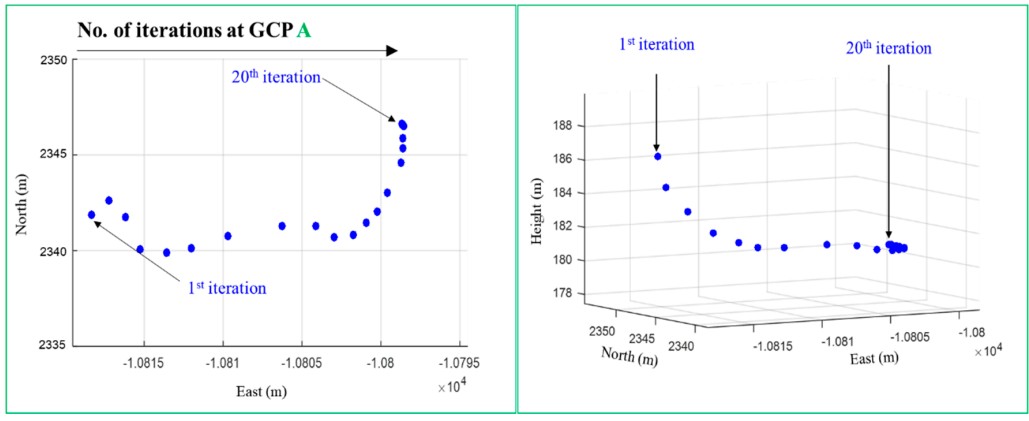

**Figure 12.** *Cont.*

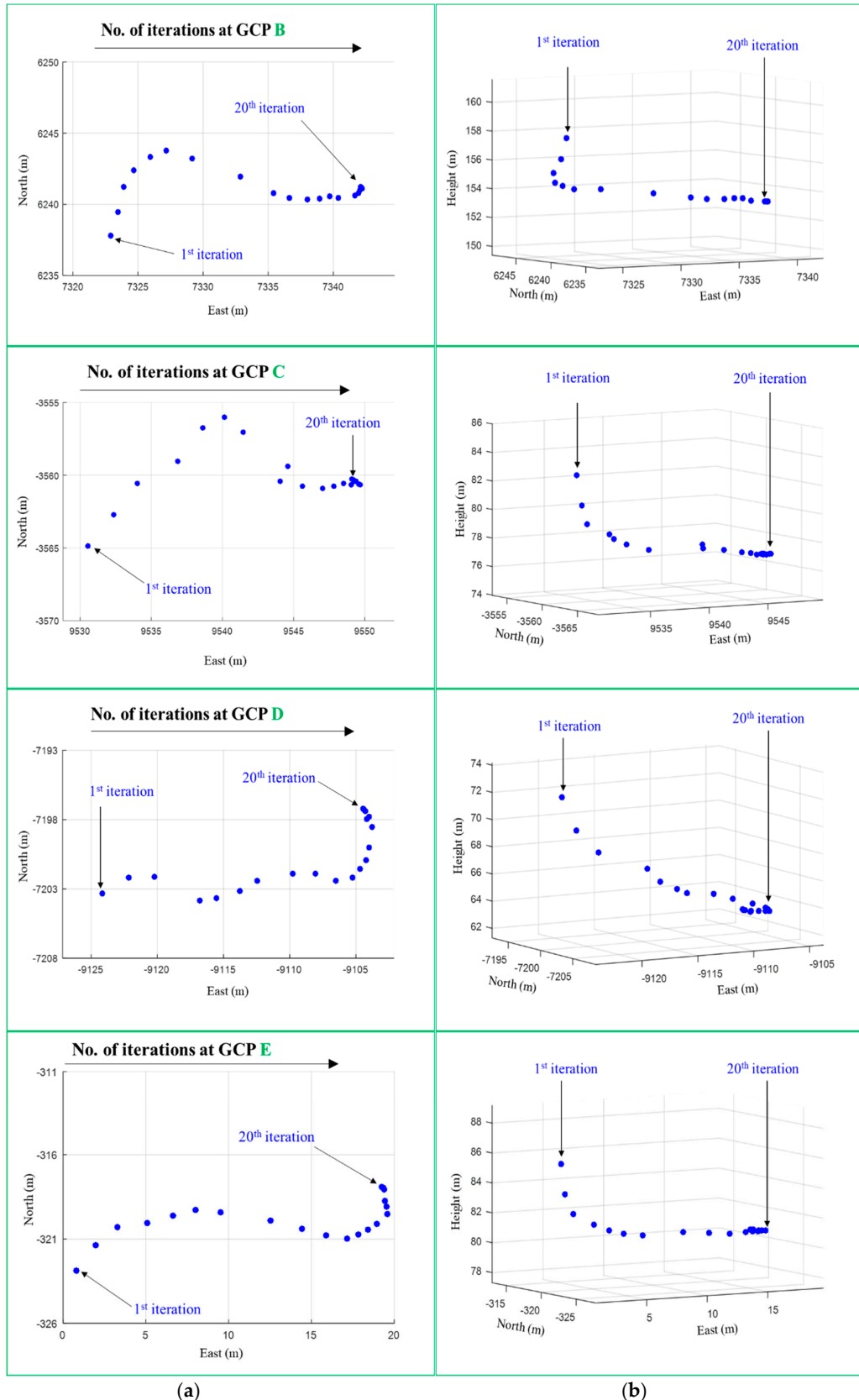

**Figure 12.** (**a**) horizontal (left column) and (**b**) vertical (right column) movement of the points over all iteration steps. For the visualization of the vertical movement, a 3D view was selected in which the influence of the horizontal component of the movement can still be seen.

### 3.2. Accuracy Investigations and Discussion of the Results

Table 2 summarizes the estimated parameters of point-to-surface matching for both cases. The horizontal shift differences are below 1 m, where the accuracies of the horizontal shifts determined by the least squares solution are approximately 2 m in both cases. The height shifts of close to 7 m differ only by 0.3 m, which corresponds well with an estimated accuracy of 0.4 m in both cases. In summary, it can be stated that the shifts resulting from point-to-surface matching differ only slightly in the two cases.

**Table 2.** Transformation parameters and its standard deviations of point-to-surface matching.

| | $X_t/\sigma_{X_t}$ | $Y_t/\sigma_{Y_t}$ | $Z_t/\sigma_{Z_t}$ | $\omega/\sigma_\omega$ | $\varphi/\sigma_\varphi$ | $\kappa/\sigma_\kappa$ |
|---|---|---|---|---|---|---|
| Case 1 | −18.5 m/2.0 m | −3.8 m/1.7 m | 7.0 m/0.4 m | N.A. | N.A. | N.A. |
| Case 2 | −19.2 m/2.0 m | −3.5 m/1.7 m | 6.7 m/0.4 m | 0.01846°/0.00580° | −0.00398°/0.00424° | −0.00585°/0.01358° |

As a reference for the point-to-surface matching results, the transformation parameter was calculated using the 15 checkpoints that were captured by GPS as well as measured in the KOMPSAT-3 images. As previously mentioned, the uncertainty in the measurement of the image coordinates leads to an accuracy of the 3D coordinates of these points of approximately 0.5 m. The estimated transformation parameters and standard deviations are listed in Table 3. The shift differences between the two cases do not exceed 0.2 m in all coordinate axes. The corresponding standard deviations $(\sigma_{X_t}, \sigma_{Y_t}, \sigma_{Z_t})$ are all approximately 0.3 m.

**Table 3.** Transformation parameters and standard deviation obtained using 15 checkpoints.

| | $X_t/\sigma_{X_t}$ | $Y_t/\sigma_{Y_t}$ | $Z_t/\sigma_{Z_t}$ | $\omega/\sigma_\omega$ | $\varphi/\sigma_\varphi$ | $\kappa/\sigma_\kappa$ |
|---|---|---|---|---|---|---|
| Case 1 | −19.2 m/0.26 m | −1.2 m/0.26 m | 5.9 m/0.26 m | N.A. | N.A. | N.A. |
| Case 2 | −19.0 m/0.29 m | −1.1 m/0.30 m | 5.8 m/0.34 m | 0.00077°/0.00542° | 0.00218°/0.00411° | 0.00351°/0.00301° |

The comparison of the results of Tables 2 and 3 is very revealing. The differences in $X_t$ of approximately 0.5 m are quite small. In $Z_t$ and $Y_t$, the differences of approximately 1 m and 2.5 m respectively are slightly larger than the standard deviations would suggest. Overall, it can be stated that the differences for horizontal shifts between the point-to-surface matching results and the results from the checkpoints reference do not exceed 3 m. Along the vertical axis, the differences are up to 1.2 m.

The estimation of the rotation parameters ($\omega, \varphi, \kappa$) leads to very small values for the matching and the reference results. They are so small that the comparison does not reveal any clear tendencies. Therefore, it can be stated that for this experiment, the rotation parameters do not contribute to the improvement of the estimation results.

Table 4 provides details how well the GPS coordinates of the checkpoints after the transformation with the point-to-surface matching parameters agree with the object coordinates of the checkpoints obtained from the image measurements. The deviations in the GPS check points without matching are visualised in Figure 7. The RMSE values show a dominant $X$-direction shift of 19.2 m, a small $Y$-direction shift of 1.3 m, and a height shift of 6.0 m. The maximum deviation of all points in all axis components is not more than 1 m from the RMSE values which indicates that the image measurements of the GPS points have been done properly. The deviations in the GPS check points after transformation with the matching parameters confirm the conclusions derived by comparing the results of Tables 2 and 3. The average differences in $X$ are below 1 m, in $Z$ below 2 m and in $Y$ below 3 m. The differences between the two cases are of minor importance.

**Table 4.** Location errors of GPS check points after point-to-surface matching.

| | without Matching | | | | after Matching | | |
|---|---|---|---|---|---|---|---|
| | X (m) | Y (m) | Z (m) | | X (m) | Y (m) | Z (m) |
| RMSE/Max | 19.2/19.9 | 1.3/2.1 | 6.0/6.7 | Case 1 | 0.8/1.5 | 2.6/3.6 | 1.1/1.8 |
| | | | | Case 2 | 0.5/1.2 | 2.6/3.6 | 1.6/3.5 |

Apparently, the RMSE values do not confirm the desired level of accuracy of half a meter, the deviations in the $Y$ component have even increased. The reason for this is expected to be less in the matching approach than in the surface model.

In Table 5, RMSE and the maximum distance values are listed between GCPs and DEM. The distance values of the GCPs to the surface at the beginning of the point-to-surface matching are 8.9 m with a maximum distance of 14.7 m. At the end of the iterations, the RMSE has reduced to 2.8 m and the maximum distance is at 8.7 m for case 1 (similar values for case 2).

**Table 5.** Distances between GCPs and DEM before and after point-to-surface matching.

| Before Matching (Iteration 0) | | | After Matching (Last Iteration) | |
|---|---|---|---|---|
| RMSE | Max | | RMSE | Max |
| 8.9 m | 14.7 m | Case 1 | 2.8 m | 8.7 m |
| | | Case 2 | 2.6 m | 8.4 m |

This signed distances are plotted in Figure 10 for all 40 TCPs and 13 UCPs. The distances across all points do not differ significantly between the results obtained with three (case 1) and with six parameters (case 2). The larger deviations can be observed for the TCPs. The left graph in Figure 10 shows the slope angles at the surface points that correspond to the TCPs and the UCPs.

The location of the points in the KOMPSAT-3 image pair were inspected, which attracted attention in Figure 10 by particularly large distances. These are mainly points located at higher altitudes in the forested area. Particularly striking are the two GCPs with the numbers 25 and 32. Figure 5 shows their location within the entire project area. The two sections of aerial photos displayed in Figure 13 suggest a rich vegetation around these two GCPs. The on-site image capture of TCP #32 (Figure 13), which is available in the NGII database, confirms this assumption. It makes it clear that bushes and trees of probably 10 m height or more surround this TCP.

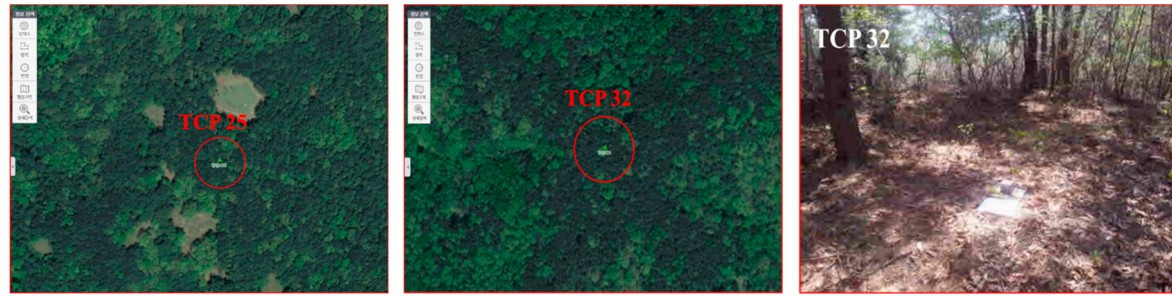

**Figure 13.** Aerial photos of two TCP locations (# 25 and #32) and field photo of TCP #32 from NGII.

The trees are the main reason why the ERDAS image matching algorithm produces a DEM that does not represent the ground in some regions, but rather approximates a surface above the ground. Since all GCPs are determined on the ground, corresponding deviations are to be expected. Due to the slope in some of the points, the vertical deviations also lead to horizontal deviations. Since the UCPs are all located in open terrain, the distances there are comparably short (cf. Figure 10).

In order to increase the accuracy achieved in the experiments, the TCPs must be carefully selected. All TCPs should be checked to see if they are in open terrain. If only those locations are included in

point-to-surface matching where the DEM exactly represents the terrain, an increase in accuracy to a level of 1 m is to be expected.

## 4. Conclusions

This paper proposes a point-to-surface matching procedure and applies it to GCPs and a DEM which is generated from KOMPSAT-3 satellite images with vendor-provided RPCs. The point-to-surface matching algorithm minimizes the sum of the squared distances between the points and the tangential planes at the correspondence points. The points used are the GCPs, more precisely the UCPs and TCPs which are provided by the Korean National Geographic Information Institute. RPC distortions are the reason why the elevation model derived from the KOMPSAT images is systematically distorted and needs to be corrected.

In the experimental investigations, the convergence behaviour of the matching algorithm was investigated as well as the achieved accuracy for correcting the DEM. The iteration behaviour of the transformed GCP shows a steady approach towards the surface. The transformed points move smoothly over all iteration steps. As the iteration number increases, the convergence manifests itself in ever smaller incremental shifts. For the given DEM and GCPs, a transformation between GCPs and the elevation model with three translation parameters was sufficient. The transformation with rotation and translation parameters has not led to any appreciable increase in accuracy. The distortion of the DEM before correction was approximately 20 m horizontally and 6 m vertically. With the proposed method, an improvement could be achieved by reducing these horizontal and vertical displacements to an accuracy level of 3 m and 2 m respectively.

An increase of this accuracy is certainly possible. For this purpose, it must be ensured that only those GCPs are included in point-to-surface matching, in which vegetation or other influences do not lead to deviations of the DEM from the terrain surface. In future work, the authors want to pursue this further. The on-site photos of the NGII are an important source for this. Alternatively, multispectral images can be used for classification, from which the type of vegetation can be derived.

**Author Contributions:** H.L. developed the idea to use GCPs (UCPs and TCPs) without measuring their image coordinates for DEM correction and implemented the algorithm. M.H. formulated the basics of the point-to-surface matching technique. Both authors carried out the experimental investigations, analysed the results and wrote the paper.

**Funding:** This work was supported by the Ministry of Education of the Republic of Korea and the National Research Foundation of Korea (NRF-2018R1D1A1B06049484).

**Acknowledgments:** The authors would like to thank the reviewers for their valuable comments.

**Conflicts of Interest:** The authors declare no conflict of interest.

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
