# Peer review of "KOMPSAT-3 Digital Elevation Model Correction Based on Point-to-Surface Matching"

_remotesensing, doi:10.3390/rs11202340_

Round 1

Reviewer 1 Report

This is an interesting paper. It is thorough and detailed in all aspects – literature references, methodology description and experimental results. A few minor suggestions:

Line 212: "Roll and yaw angles of both images point in a similar direction, the yaw angles differ essentially in their sign." – The second “yaw” here should be “pitch”

UCPs and TCPs – should define them for the benefit of readers unfamiliar with these terms. What does unified mean in the UCPs? What is "triangulation" in TCP for?

Lines 190-194: "If the horizontal displacement between the DEM and the GCPs is a multiple of the mesh size of the DEM, this will cause the transformed GCPs to move across several meshes within the iterative solution of the matching problem. The normal vector may differ significantly between adjacent meshes. In order to mitigate this influence, only one third of the estimated values are taken into account when updating the parameters." This is not very clear. Does the first sentence imply that the horizontal displacement is greater than the mesh size (rather than an integral multiple of mesh size)? Why will transformed GCPs move across the mesh a distance of several units (I assume that is what is meant here, rather than several meshes)? How does taking one third of the estimated values mitigate the influence? What does one-third mean here? Are you using one-third of the estimated horizontal displacement for each of the GCPs or are you using one-third of the number of estimated displacements (i.e., if there are 15 GCPs, are you using displacements estimated for 5 of them)?

Levenberg-Marquardt algorithm: Need to provide a reference

Author Response

We appreciate the constructive criticism and the valuable comments that we have included in the revised manuscript. In the following, we first repeat the suggestions, comments and wishes of the reviewer. Our explanations are then given in blue writing.

Reviewer 1

This is an interesting paper. It is thorough and detailed in all aspects – literature references, methodology description and experimental results. A few minor suggestions:

Line 212: "Roll and yaw angles of both images point in a similar direction, the yaw angles differ essentially in their sign." – The second “yaw” here should be “pitch”

- This is correct. We corrected the typo.

UCPs and TCPs – should define them for the benefit of readers unfamiliar with these terms. What does unified mean in the UCPs? What is "triangulation" in TCP for?

- We have added a short explanation of the background of UCP and TCP in lines 86 ff. of the manuscript.

Lines 190-194: "If the horizontal displacement between the DEM and the GCPs is a multiple of the mesh size of the DEM, this will cause the transformed GCPs to move across several meshes within the iterative solution of the matching problem. The normal vector may differ significantly between adjacent meshes. In order to mitigate this influence, only one third of the estimated values are taken into account when updating the parameters." This is not very clear. Does the first sentence imply that the horizontal displacement is greater than the mesh size (rather than an integral multiple of mesh size)? Why will transformed GCPs move across the mesh a distance of several units (I assume that is what is meant here, rather than several meshes)? How does taking one third of the estimated values mitigate the influence? What does one-third mean here? Are you using one-third of the estimated horizontal displacement for each of the GCPs or are you using one-third of the number of estimated displacements (i.e., if there are 15 GCPs, are you using displacements estimated for 5 of them)?

- We have now formulated misleading wording more precisely. In addition, we have added a new figure (Fig. 11) in Section 3, which illustrates the iterative behavior in relation to the transformed GCPs.

Levenberg-Marquardt algorithm: Need to provide a reference

- A reference has been added to the literature.

Reviewer 2 Report

In this paper, a point-to-surface matching is presented to produce high-quality DEM from KOMPSAT-3 images with existing GCPs. In the proposed point-to-surface matching, the transformation between exiting GCPs and the surface was approximated using an iterative least squares solution. The horizontal and vertical accuracy of the propose method were evaluated with two test cases: with and without rotation. In general, the method is presented with clarity and in detail and was sufficiently tested. Please address the following minor comments:

The literature review does not sufficiently reflect the latest development in the field. Most of the references were published before 2011. Line 105: “Instead of the height difference we use the shortest distance from the point”.

Please provide justification for using the shortest distance from the point instead of the height difference.

Line 204: “NGII website.”

Please specifiy the web address of the NGII website.

Line 211: “Yangsan city region.”

Please be more specific about where Yangsan city is located in South Korea.

Line 219-220: Table 1

The column header “Sensor” does not seem to well capture the parameter information in the column. Specifically, “Acquisition date/time” is not a sensor property.

Line 244-246: Figure 7.

Please indicate the unit for each axis in both the left and right figures.

Line 280: “The change the normal vectors…”

Please rephrase or correct the sentence.

Line 283: “During the first 15 iterations numerous changes”

“Numerous changes” seems confusing.  Please rephrase or correct.

Author Response

We appreciate the constructive criticism and the valuable comments that we have included in the revised manuscript. In the following, we first repeat the suggestions, comments and wishes of the reviewer. Our explanations are then given in blue writing.

Reviewer 2
In this paper, a point-to-surface matching is presented to produce high-quality DEM from KOMPSAT-3 images with existing GCPs. In the proposed point-to-surface matching, the transformation between exiting GCPs and the surface was approximated using an iterative least squares solution. The horizontal and vertical accuracy of the propose method were evaluated with two test cases: with and without rotation. In general, the method is presented with clarity and in detail and was sufficiently tested. Please address the following minor comments:

The literature review does not sufficiently reflect the latest development in the field. Most of the references were published before 2011.

- There are actually three temporal phases in the development of procedures of matching points to elevation models. The early phase dates back to the time around 1990 when the developments began in photogrammetry. From around 2005 to 2011 extensions were presented (e.g. by Grün and co-workers) and remote sensing specific features were researched. In the last five years, applications with different remote sensing sensors have become more and more of a focus. To discuss the many experimental investigations on nearly the same theoretical basis (e.g. the LHD alg.) we considered as not so helpful. With recent work Photogrammetry/Computer Vision we could extent towards coarse/fine registration, features (lines, planes) extracted from point clouds and used for matching, semantic features, etc., discussed e.g. in

Xu, Y., Boerner, R., Yao, W., Hoegner, L. and Stilla, U., 2017.
Automated coarse registration of point clouds in 3d urban scenes using voxel based plane constraint. ISPRS Annals of the Photogrammetry, Remote Sensing and Spatial Information Sciences IV-2/W4, pp. 185–191. Yang, B. and Zang, Y., 2014. Automated registration of dense terrestrial laser-scanning point clouds using curves. ISPRS Journal of Photogrammetry and Remote Sensing 95, pp. 109–121. Yang, B., Dong, Z., Liang, F. and Liu, Y., 2016. Automatic registration of large-scale urban scene point clouds based on semantic feature points. ISPRS Journal of Photogrammetry and Remote Sensing 113, pp. 43–58.

- It would be easy to mention them in the literature review. But there are no consequences (neither theoretical nor practical) for the work in our paper. For this reason, we have refrained from expanding the literature review further.

Line 105: “Instead of the height difference we use the shortest distance from the point”.
Please provide justification for using the shortest distance from the point instead of the height difference.

- We have reformulated the sentence (line 105) in such a way that it becomes clear that it is not a statement but a target for the subsequent equations and statements.

Line 204: “NGII website.”
Please specifiy the web address of the NGII website.

- We added the web address.

Line 211: “Yangsan city region.”
Please be more specific about where Yangsan city is located in South Korea.

- In Figure 4 we have added a map of the urban region of Yangsan on the Korean Peninsula.

Line 219-220: Table 1
The column header “Sensor” does not seem to well capture the parameter information in the column. Specifically, “Acquisition date/time” is not a sensor property.

- We deleted the term “Sensor” in Table 1.

Line 244-246: Figure 7.
Please indicate the unit for each axis in both the left and right figures.

- The units for all axes are added in Figure 7.

Line 280: “The change the normal vectors…”
Please rephrase or correct the sentence.

- is corrected

Line 283: “During the first 15 iterations numerous changes”
“Numerous changes” seems confusing.  Please rephrase or correct.

- is formulated more precisely

Reviewer 3 Report

This article is interesting. Some issues that could be fixed.

In introduction please check Your references, they suppose to be in numerical order, while you have 1,2,3,4 and then 14, 15.

In line 35 letter S is missing in word ample.

In lines from 211 to 215 please avoid use of word about, there are precise values of image size and resolution - use those.

It would be useful for readers outside the South Korea to know where the study area is. Recommendation for Figure 4 - add one image where You show Korean peninsula with red dot labeling Yangsan area.

Line 239 - 240, why are the GPS control points observed in 2013 and not in 2019. Please explain.

Figure 7 right image is missing unit label

Figures 8, 9 and 10 have poor resolution, please improve if possible.

In Conclusion section please mention what would Your future work include.

Author Response

We appreciate the constructive criticism and the valuable comments that we have included in the revised manuscript. In the following, we first repeat the suggestions, comments and wishes of the reviewer. Our explanations are then given in blue writing.

Reviewer 3
This article is interesting. Some issues that could be fixed.
In introduction please check Your references, they suppose to be in numerical order, while you have 1,2,3,4 and then 14, 15.

- We have modified it. The numerical order of all references is now realized.

In line 35 letter S is missing in word ample.

- ample means “enough, sufficient, adequate, satisfactory” which fits well to line 35: “there is also ample experimental evidence for KOMPSAT imagery that relative accuracy to meter level is attainable”.

In lines from 211 to 215 please avoid use of word about, there are precise values of image size and resolution - use those.

- We have removed the “about” for the resolution of the images and gave the precise numbers. With the generalizing indication of an area of approx. 20 km x 16 km in the urban region Yangsan we want to express the size of the project area, which is covered by the two KOMPSAT-3 images. By multiplying the image size with the resolution the “exact” numbers for the ground size of each image can be calculated. However, these numbers do not have any added value.

It would be useful for readers outside the South Korea to know where the study area is. Recommendation for Figure 4 - add one image where You show Korean peninsula with red dot labeling Yangsan area.

- We have taken up this suggestion and supplemented Figure 4 with a map showing the Korean peninsula and the Yangsan city region.

Line 239 - 240, why are the GPS control points observed in 2013 and not in 2019. Please explain.

- The acquisition of the KOMPSAT images was in January 2013. The GPS measurement took place only three months later, so that the data should fit together well due to the small time difference.

Figure 7 right image is missing unit label

- We added the unit [m] in Figure 7.

Figures 8, 9 and 10 have poor resolution, please improve if possible.

- In the Word document, the images are scaled to 57%. At 100% they are still very sharp. So we suspect that the version viewed by the reviewer may have been compressed?

In Conclusion section please mention what would Your future work include.

- We have added an idea that we want to pursue in the future.
